# The Role of Artificial Intelligence in Decoding Speech from EEG Signals: A Scoping Review

**DOI:** 10.3390/s22186975

**Published:** 2022-09-15

**Authors:** Uzair Shah, Mahmood Alzubaidi, Farida Mohsen, Alaa Abd-Alrazaq, Tanvir Alam, Mowafa Househ

**Affiliations:** 1College of Science and Engineering, Hamad Bin Khalifa University, Doha P.O. Box 34110, Qatar; 2AI Center for Precision Health, Weill Cornell Medicine-Qatar, Doha P.O. Box 34110, Qatar

**Keywords:** sensors, speech decoding, electroencephalogram, signals, artificial intelligence, imagine speech

## Abstract

*Background:* Brain traumas, mental disorders, and vocal abuse can result in permanent or temporary speech impairment, significantly impairing one’s quality of life and occasionally resulting in social isolation. Brain–computer interfaces (BCI) can support people who have issues with their speech or who have been paralyzed to communicate with their surroundings via brain signals. Therefore, EEG signal-based BCI has received significant attention in the last two decades for multiple reasons: (i) clinical research has capitulated detailed knowledge of EEG signals, (ii) inexpensive EEG devices, and (iii) its application in medical and social fields. *Objective*: This study explores the existing literature and summarizes EEG data acquisition, feature extraction, and artificial intelligence (AI) techniques for decoding speech from brain signals. *Method*: We followed the PRISMA-ScR guidelines to conduct this scoping review. We searched six electronic databases: PubMed, IEEE Xplore, the ACM Digital Library, Scopus, arXiv, and Google Scholar. We carefully selected search terms based on target intervention (i.e., imagined speech and AI) and target data (EEG signals), and some of the search terms were derived from previous reviews. The study selection process was carried out in three phases: study identification, study selection, and data extraction. Two reviewers independently carried out study selection and data extraction. A narrative approach was adopted to synthesize the extracted data. *Results*: A total of 263 studies were evaluated; however, 34 met the eligibility criteria for inclusion in this review. We found 64-electrode EEG signal devices to be the most widely used in the included studies. The most common signal normalization and feature extractions in the included studies were the bandpass filter and wavelet-based feature extraction. We categorized the studies based on AI techniques, such as machine learning and deep learning. The most prominent ML algorithm was a support vector machine, and the DL algorithm was a convolutional neural network. *Conclusions*: EEG signal-based BCI is a viable technology that can enable people with severe or temporal voice impairment to communicate to the world directly from their brain. However, the development of BCI technology is still in its infancy.

## 1. Introduction

Speech is a fundamental requirement of daily life and the primary mechanism of social communication. Certain mental disorders, diseases, accidents, vocal abuse, and brain traumas can result in permanent or temporary speech impairment, significantly impairing one’s quality of life and occasionally resulting in social isolation [1]. As a result, Brain–computer interfaces (BCIs) have been developed to enable people with speech issues and paralyzed patients to communicate immediately with their surroundings. BCI is a computer-based system that detects, analyzes, and converts brain signals into commands delivered to an output device to perform the desired action [2]. It has a lot of applications, such as communication systems and controlling external devices. BCI systems have traditionally been used for seriously impaired persons, but healthy people now use them for communication and everyday life assistance systems [2].

Exogenous and endogenous paradigms are two approaches in the BCI paradigm for brain signal generation [3]. The former measures brain waves responding to an external stimulus [4], while the latter utilizes mental tasks, such as motor imagery (MI) and imagined speech (IS), while recording brain signals [5,6]. External devices are not required in the endogenous paradigm; as a result, their application in BCI has been growing recently [7]. Traditionally, MI is the most researched paradigm; it measures the waves generated by our brain as we imagine ourselves moving [7]. Thus, the number of classes is limited due to the imaginative nature of body movement (i.e., the right hand, the left hand, both hands, and feet). While MI has improved patients’ communication ability, its practical application is limited because of the low number of classes and the requirement of external stimuli [8]. Therefore, intuitive paradigms are becoming prominent for solving these problems, such as IS.

Recently, IS, active thought, and covert speech, have been investigated as an intuitive paradigm [9]. It is a process by which a person attempts to imagine pronouncing a word without moving the articulatory muscles or making an audible sound. As a result of its intuitiveness, this paradigm is suited for developing communication systems. IS has multiclass scalability [10], which shows the potential of constructing an expandable BCI system. For IS data acquisition, invasive and non-invasive methods are being used. Recent studies have investigated various technologies for recording the brain signals related to speech imagination. Among these technologies are electroencephalography (EEG), which records the electrical activity of the human brain; electromyography (EMG), which records articulatory and facial muscle movement; and invasive electrocorticography (ECoG), which records electrical activity on the brain’s surface. However, due to the high temporal resolution, mobility, and low cost, EEG-based technologies are more common than others for BCI systems. This scoping review is particularly interested in decoding speech from EEG signals.

IS decoding methods have relied on classification algorithms and feature extraction; therefore, several works have been published in either the direction of extracting appropriate features, developing optimal classification models, or both. Spectro-temporal features [11], common spatial patterns [12], and autoregressive coefficients [13] are some of the features that are used to represent IS in EEG. Different conventional machine learning algorithms have been used to decode IS from EEG, such as random forests [14], support vector machines (SVMs) [7,15], k-nearest neighbors [16], linear discriminant analysis (LDA) [13,17], and naive Bayes [13]. Recently, many studies have been using deep learning methods to automatically extract speech features to improve performance [18,19,20]. Convolutional neural networks (CNN) are the most widely used deep learning approach concerning BCI/EEG.

Previous reviews on decoding speech from EEG signals using artificial intelligence (AI) techniques have not been thoroughly conducted. Early reviews have attempted to summarize the use of AI for decoding speech. The review [21] focuses on the IS and searches PubMed and IEEE Xplore for relevant studies. However, we focus on covert and overt speech and search six different databases, including medical and technical. The reviews [22,23] explored different data modalities used for speech recognition from neural signals and do not provide an overview of other AI and feature extraction techniques used in the literature. However, we summarized different AI and feature extraction techniques in this study. Therefore, to fulfill the gap in the existing review, we conducted a scoping review to explore the role of AI in decoding speech from brain signals.

## 2. Method

A scoping review was conducted to explore the available research literature and address the aforementioned issues. We followed the guidelines of the PRISMA-ScR extension of the scoping review for this review [24]. In the following subsection, we explain the details of the method adopted for carrying out this scoping review.

### 2.1. Search Strategy

#### 2.1.1. Search Sources

In the current studies, we searched the following electronic databases: PubMed, IEEE explorer, the ACM Digital Library, Scopus, arXiv, and Google Scholar. We selected the first 100 citations sorted by relevance from Google Scholar, as Google Scholar retrieved several thousand studies. The search process was carried out from 23 to 25 October 2021.

#### 2.1.2. Search Terms

We considered search terms based on three different elements: intervention (imagined speech, covert speech, active thoughts, inner speech, silent speech), datatype (electroencephalography, electropalatographic, and electromyography), and intervention (encoding decoding, artificial intelligence, deep learning, machine learning, and neural network). These search terms were sufficient to retrieve all relevant studies. Some of these search terms were derived from previous reviews. Further, we used different search strings in each electronic database because the search process and length of the search strings are different in other databases. The details of the search strings with the number of citations retrieved for searching each database were captured (available in Appendix A).

### 2.2. Study Eligibility Criteria

The primary focus of this review is to explore the AI technology or approaches used to decode human speech from electroencephalography (EEG) signals. The studies included in this review should meet the following eligibility criteria:(1)AI-based approach used for decoding speech.(2)EEG signal data used to build the AI model.(3)Prediction labels/classes must be words or sentences.(4)Written in English.(5)Published between 1 January 2000 to 23 October 2021.(6)Peer-reviewed articles, conference proceedings, dissertations, thesis, and preprints were included, and conference abstracts, reviews, overviews, and proposals were excluded.

No constraints were imposed on study design, country of publication, comparator, or outcome.

### 2.3. Study Selection

The study selection process was carried out in three steps: In the first step, duplicate studies were identified using the auto-duplicated detection function of the Rayyan software [25] and were removed. In the second step, two reviewers (US and MA) screened unique titles and abstracts. In the final step, reviewers read the full text of the included studies. Any disagreement between reviewers was resolved through discussion.

### 2.4. Data Extraction and Data Synthesis

Before extracting data from the included studies, we designed an extraction table using an Excel spreadsheet and had it verified by AAA and MH. The pilot test included two studies to ensure the consistency and availability of data. We extracted data related to the study characteristics (i.e., type of publication, year of publication, and country of publication), attributes of AI technologies (i.e., AI branch, specific algorithm, AI framework, programming language, performance metrics, loss functions, etc.), and characteristics related to data acquisitions (i.e., dataset availability, subject conditions, the toolbox for recording, type of speech, language of the dataset, number of electrodes, etc.). The extracted data are summarized in Tables. We conducted a narrative approach to synthesize our results and findings.

## 3. Results

### 3.1. Search Results

A total of 263 citations were retrieved by searching six electronic databases, as demonstrated in Figure 1. We identified 84 duplicate studies and removed them from the corpus. A total of 179 studies were passed to the title and abstract screening phase, from which 135 citations were excluded for various reasons. In the final step, we read the full text of the 44 articles and excluded ten studies. A total of 34 studies were included in this review.

### 3.2. Studies’ Characteristics

In the included studies, about (44%) of the studies were peer-reviewed articles (n = 15) [2,3,4,5,6,7,8,10,11,12,14,15,16,18,26], whereas 47% of the studies were conference proceedings (n = 16) [9,13,17,19,20,27,28,29,30,31,32,33,34,35,36,37] and around 9% were preprint articles (n = 3) [38,39,40]. These articles were published in 15 different countries. Most of the studies were published in Korea (n = 8) [4,7,14,19,26,36,37,40] and India (n = 6) [2,9,15,16,30,31], as presented in Table 1. The greatest number of papers were published in 2020 and 2021. The least number of studies were published from 2013 to 2016.

### 3.3. AI-Enabled Technique for Speech Decoding from EEG Signals

Table 2 summarized the characteristics of the AI-enabled techniques used for decoding speech (can be covert or overt speech) from EEG signals. Of 34 studies, six articles (n = 6) [4,9,19,20,28,30] used both machine learning (ML) and deep learning (DL) in their experiments, whereas ML-based techniques were used in 15 (44%) [2,5,7,8,11,12,13,14,15,16,17,26,34,35,36] studies, and DL-based techniques were used in 13 studies (40%) [3,6,10,18,27,29,31,32,33,37,38,39,40]. However, the support vector machine (n = 10), linear discriminant analysis (n = 9), and k-nearest neighbor (n = 7) algorithms were the most frequently used ML techniques. On the other side, the convolutional neural network (n = 7) [10,18,29,33,37,39,40], recurrent neural network (n = 4) [27,32,38,39], and deep neural network (n = 3) [3,6,31] algorithms were the most frequently used DL-based techniques in the included studies. Some of the included studies used more than one algorithm in their experiments. Therefore, we show only the frequently used algorithms in Table 2.

Most of the included studies did not mention their AI framework and programming languages explicitly. However, seven studies [9,18,27,30,33,38,39] used a TensorFlow framework, studies [9,17] used Scikit learn, and [20] used PyTorch. Out of 34 studies, eight studies [9,18,20,27,33,38,39,40] and six studies [7,12,13,14,28,40] utilized Python and the MATLAB programming language, respectively. Around 60% of studies were limited in revealing the detail of the programming language used in the study.

To measure the performance of the AI models in the included studies, 29 studies used accuracy [2,3,5,6,7,8,9,10,11,12,13,14,15,17,18,19,20,26,28,29,30,32,33,34,35,36,37,40], 5 used the kappa score [12,14,15,35,38], and 4 used word error rates [16,27,38,39] as the performance metric. However, some studies used more than one performance metric to estimate the performance of the model. Nevertheless, 21 of the included studies used a K-fold cross validation [2,4,5,6,7,10,11,12,13,14,15,16,19,20,26,28,29,31,35,37,40] and 10 used a train test split [3,9,17,18,27,30,32,34,38,39] for model validation.

A total of 17 of the included studies reported their loss function. The cross entropy [3,6,9,18,20,27,31,32,37] and mean square error [29,30,39] are widely used as the loss function.

### 3.4. Characteristics of Dataset

Of the included studies, 21 created their dataset for AI model training and testing, while 13 studies utilized publicly available datasets. The dataset size was reported in 30 of the included studies and ranged from 200 to 14,400 trials. A data sample size of less than 1000 was reported in more than half of the included studies (n = 18), and a sample size greater than 1000 and less than 2500 was reported in seven studies. The sample size equal to or greater than 2500 was reported in four studies.

Bandpass filtering techniques were the most common signal normalization technique reported in 31 studies. However, after bandpass filtering, nine studies used an independent component analysis to remove the artifact from the signals, and eight studies used notch filter. A bandpass filter with a Kernel principal component analysis, min–max scaling, common average reference, and cropped decoding technique and frequency-specific spatial filters were used in one study each.

One of the challenging processes for decoding speech from raw EEG signals is feature extraction. It has a significant impact on the performance of the model. If one can extract meaningful and relevant features from raw EEG signals, then there is a greater chance that the model will perform better during discrimination between different classes. Therefore, the details of the feature extraction were reported in all included studies. Wavelet-based feature extraction techniques were reported in seven studies, followed by five simple features in five studies and common spatial patterns in five studies (Table 3).

There were 15 different feature extraction techniques reported in the included studies. The common spatial pattern feature extraction technique was reported in six studies, followed by simple features (i.e., min, max, average, std, etc.), which was written in five studies. Discrete wavelet transform was reported in five studies, as demonstrated in Table 2.

The training set portion of the total dataset was reported in 23 studies. A training set size ≥90% was reported in 11 studies, ≥80% was reported in seven studies, ≥70% was reported in two studies, and <70% was reported in three studies.

The testing set portion of the total dataset was detailed in 17 studies. A test set size ≥30% was documented in two studies, ≥20% was documented in eight studies, and <20% was documented in seven studies. The validation set portion of the total dataset was revealed in only seven studies. A validation set size of ≥20% was reported in two studies, while <20% was detailed in five studies.

### 3.5. Characteristics of Data Acquisition from EEG Signal

The 27 included studies revealed that the dataset was collected from normal and healthy subjects. However, eight studies did not report the condition of the subjects.

The speech language used to record the EEG signals was reported in all of the included studies. More than three-quarters (n = 27) of the dataset was recorded in English, followed by Spanish in four studies, as shown in Table 4. Dataset classes were words (n = 31) and sentences (n = 4) in the included studies.

In the Introduction, we the types of speech (covert and overt). In 22 studies, the EEG signals were recorded, while one imagined words/sentences in their mind. Fewer studies recorded the EEG signal of overt speech. However, 11 studies recorded the EEG signals of both overt and covert speech.

Thirty-four studies reported the stimulus (visual/audio) upon capturing the EEG signals. However, 33 studies used visual or audio stimuli to record the EEG signals. One of the studies recorded the EEG signals without providing any stimulus to the participants.

Thirteen studies revealed information about background noise during the recording of the EEG signals. Out of 13 studies, 8 recorded the data over some background noise, while 5 isolated the participant in a silent room to record the signals.

Initial rest impacts capturing the subject’s data, specifically the EEG signals, because body and eye movement and active thinking affect the EEG signal. Twenty-two of the included studies report the initial rest. However, the rest time is different in different studies, so we categorized it into three classes: initial rest ≥ 5 s was reported in nine studies, 3–4 s was reported in 4 studies, and 1–2 s was reported in nine studies.

The rest between each trial also has an impact on the data’s consistency. Out of 34 studies, 24 reported details about rest between 1–2 s; rest between trials was reported in nine studies, ≥5 s was reported in eight studies, and 3–4 s was reported in seven studies.

Some EEG device companies provide their toolbox for recording brain signals, such as NeuroScan. On the other hand, some free available toolboxes are compatible with various EEG devices and are commonly used, such as EEGLAB. Of the included studies, 21 reported the toolbox for recording EEG signals. About half of the studies utilized EEGLAB. E-Prime and OpenBMI were used in two studies, and NeuroScan and PyEEG toolboxes were used in one study. The details of the devices were reported in 23 studies. Of those studies, ten of the studies used the Brian EEG device for EEG recording, eight studies used the NeuroScan EEG device, three studies used Emotive, one study used OpenBCI, and one study used Biosemi ActiveTwo.

Of 34 studies, 31 reported the number of electrodes used to capture the brain signal of the participants. Of those 31 studies, 20 studies used a 64-electrode cap, 6 studies utilized less than 32 electrodes, 4 studies used 32 electrodes, and 1 study used 128 electrodes to record the EEG of the subject. Of the included studies, ten studies reported the setting of data acquisition. Among ten studies, five studies recorded data in the office, followed by three studies in the lab, and two studies in an isolated room.

## 4. Discussion

### 4.1. Principal Findings

In this study, we carried out a scoping review of the use of AI for decoding speech from human brain signals. We limit the scope of this review to only the data modality of EEG and word and sentence prompts. Conversely, previous studies have focused on different data modalities, such as ECoG, fMRI, and fNIRS, while some focus on syllables [11,41,42] and motor imagery [43,44]. We mainly categorized the studies based on AI branches, such as machine learning and deep learning, as presented in Table 2. Support vector machines and linear discriminant analysis algorithms were the most used machine learning techniques.

Conversely, convolutional neural and artificial neural network algorithms were widely used in the deep learning framework. Few studies have utilized both machine learning and deep learning algorithms. The k-fold cross-validation technique was used in numerous studies irrespective of the AI branch. Out of 34 studies, 29 used accuracy to measure model performance because most of the datasets were balanced, and they used their private datasets.

EEG signals are very noisy; proper normalization is required before feature extraction and feeding to AI models. Bandpass filtering techniques were used in each study, combined with other normalization techniques. The choice of low pass and high pass varies in the included studies. Normalization techniques, such as min–max scaling and Kernel principal component analysis, were used in a few studies. To train the model, handcrafted features or feature engineering are required to extract meaningful features from raw EEG signals. The model’s performance heavily depends on the features extracted from the raw data. Common spatial patterns, simple features (i.e., min, max, average, std, var, etc.), and discrete wavelet transformations were mainly utilized as feature extraction techniques. These kinds of feature extraction techniques were also highlighted in another review [21].

Most of the studies used 64-channel EEG devices to capture the brain signals of the subjects. However, only one study used a 128-channel EEG device to record the brain signals of the subjects. A few studies used 32-channel or less than 32-channel devices. Another review conducted by Panachakel et al. [21] also reported that a 64-channel was the most used for recording EEG signals for speech decoding.

### 4.2. Practical and Research Implications

#### 4.2.1. Practical Implications

We have summarized a list of different AI and feature extraction techniques for decoding speech directly from human EEG signals. Brain–computer interface developers can use this list to develop a reliable and scalable BCI system based on speech and identify the most appropriate AI and feature technique for speech decoding.

The current review identifies relatively few included studies that focus on decoding continuous words or sentences from EEG signals despite its healthcare advantages. We recommend BCI system designers and healthcare professionals consider continuous word prediction from EEG signals. Continuous word prediction can be regarded as mind reading and enable people with cognitive disabilities to communicate with the external environment. This indicates that a continuous word prediction system can be more feasible for users who suffer from a severe cognitive disability and voice impairment.

Most of the included studies used accuracy to measure their models’ performance. A few studies used other performance metrics, such as precision, recall, and word error rates. We recommend that the BCI developer use other performance metrics, such as AUC, sensitivity, specificity, and mean square, to provide a more holistic view of a model’s performance. This will allow us to understand how the model is sensitive to different classes. Conversely, most of the included studies used the k-fold cross-validation technique to validate and generalize the model. However, none of the included studies used external validation techniques to validate and generalize the model performance. Therefore, we recommend that the research community and developer use external validation to provide a holistic and generalized picture of the model in real-time.

Of the included studies, six utilized machine learning and deep learning techniques. Among those studies, two studies utilized deep learning for extracting features and used machine learning for classification. For instance, [19] used a Siamese neural network to extract features from the EEG signals and used a K-NN to predict the word. Alternatively, the studies [9,19,28,30] compared machine learning techniques with deep learning techniques and revealed that deep learning is prominent in classifying EEG signals.

Considering the machine learning-based modeling technique, SVM and LDA were reported in most studies. None of the included studies used a gradient boosting machine (GBM) algorithm, as GBM can learn the complex hidden pattern from EEG signals’ data and convert simple and weak learners into better learners. We recommend that the research community use GBM algorithms, such XGBoost and AdaBoost, etc. On the other hand, convolutional neural networks and artificial neural networks were the most commonly used deep learning algorithms. Relatively few studies utilized a recurrent neural network (RNN) for speech decoding, as an RNN may better learn temporal and sequence information from the EEG signals [45,46]. We believe that the use of an RNN could improve the performance of the models in speech decoding and speech recognition from the EEG signals’ data.

#### 4.2.2. Research Implications

The scope of this review is to summarize different AI and feature extraction techniques reported in the literature for decoding speech directly from the brain using EEG signals. Future reviews that link the different data acquisition protocols used to capture the EEG signals of a subject for decoding speech are possible and recommended. Of the included studies, the dataset size of less than 1000 samples, reported in more than half of the studies as the model performance, is directly related to the dataset size. We can achieve a better model performance on large datasets. Therefore, we recommend preparing large datasets for future use. Very few publicly available datasets of EEG signals for speech decoding were noted in the existing literature, given that there are privacy and security concerns when publishing any dataset online. However, we recommend that the research community in the field de-identify the dataset and make it available for other researchers to develop new AI and feature extraction techniques.

Relatively few studies have focused on and developed AI models for continuous word/sentence prediction from direct brain electrical signals. Continuous word prediction can be considered mind reading and serve to enable people with cognitive disabilities and voice impairment to communicate with others. Therefore, further research would require focusing on the continuous word prediction system from EEG signals. Moreover, we limited the scope of this review to word or sentence prompts and excluded studies that focus on syllables or vowels. We recommend that future reviews focus on the syllable and vowels to summarize other feature extraction and AI techniques in the field.

We explicitly focus on the EEG data modality because it is a non-invasive way to measure the brain activity of humans by placing EEG electrodes on the head of the subjects. EEG signals have high temporal resolution even though the spatial resolution is poor. In the literature, other data modalities are used for decoding speech from the brain, such as ECoG, fMRI, fNIRS, etc. Conversely, a future review could focus on analyzing other data modalities.

### 4.3. Strengths and Limitations

#### 4.3.1. Strengths

This review summarizes the current literature for decoding speech from EEG signals. We shed light on different AI techniques, feature extraction, signal filtering, normalization, and data acquisition used to decode speech from brain activity. We used six comprehensive databases to obtain the most relevant and up-to-date studies. To the best of our knowledge, this is the first review to cover IS detection based on EEG signals and to focus on AI techniques. Therefore, this review will benefit interested researchers and developers in this domain.

#### 4.3.2. Limitations

We limited the scope of this review to isolated word and sentence prediction from EEG signals; therefore, we may have missed significant studies. Moreover, we focused on the data modality of EEG signals. We excluded other modalities, such as ECoG, FNIR, fMRI, etc., so we may have missed some studies that used EEG signals and other modalities. We obtained a sufficient number of studies; therefore, we did not perform a backward and forward reference checklist. We only considered studies published in English, so we may have overlooked some significant studies published in other languages.

## 5. Conclusions

We have summarized a list of AI-based and feature extraction techniques for decoding speech from human EEG signals. Developers of brain–computer interfaces can use this list for a reliable and scalable BCI-based system and identify the most appropriate AI and feature technique for speech decoding. In spite of its health benefits, relatively few studies have investigated the decoding of continuous words or phrases from EEG signals. We recommend BCI system designers and healthcare professionals consider continuous word prediction from EEG signals. Continuous word predictions can be regarded as mind reading and allow people with cognitive impairments to communicate with the external environment. This indicates that the continuous word prediction system may be more feasible for users with severe cognitive or voice impairments.

## Figures and Tables

**Figure 1 sensors-22-06975-f001:**
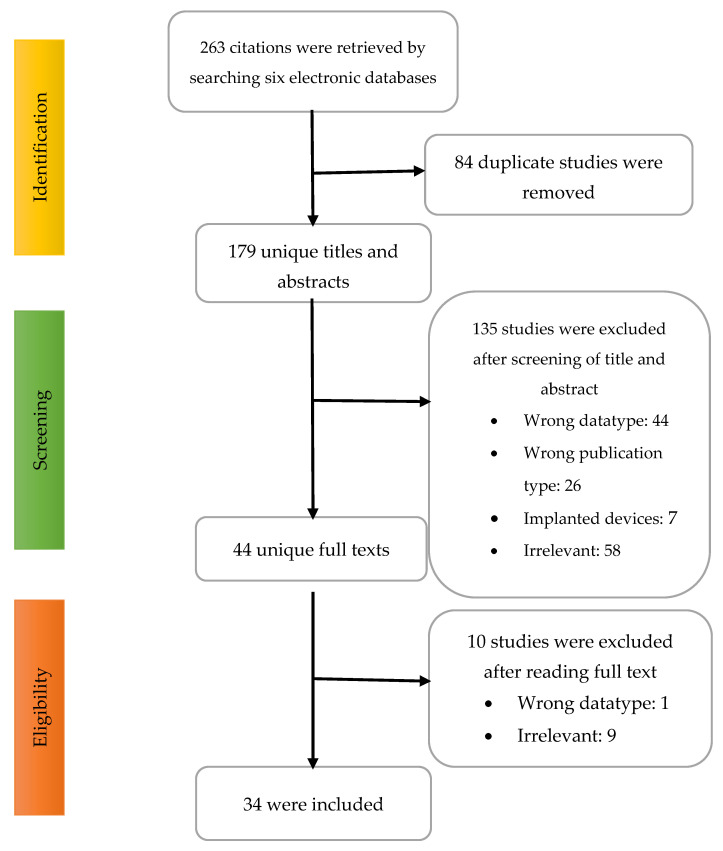
Study selection flow chart.

**Table 1 sensors-22-06975-t001:** Studies’ characteristics.

Characteristics		Studies (n)
Publication type		
	Peer-reviewed article	15
	Conference paper	16
	Preprint	3
Year of publication		
	2021	6
	2020	12
	2019	5
	2018	2
	2017	5
	2016	2
	2015	1
	2013	1
Country of publication		
	Korea	8
	India	6
	United States	4
	Canada	3
	United Kingdom	3
	Iran	2
	Bangladesh	1
	China	1
	Slovakia	1
	Malaysia	1
	Mexico	1
	Saudi	1
	Rumania	1
	Columbia	1

**Table 2 sensors-22-06975-t002:** Characteristics of AI-based techniques.

Characteristics		Study n
AI branch		
	Machine learning (ML)	15
	Deep learning (DL)	13
	Both (ML & DL)	6
Algorithm		
	Support vector machine	10
	Linear discriminant analysis	9
	K-nearest neighbor	7
	Random forest	5
	Decision tree	2
	Naive Bayes	2
	Convolutional neural network	9
	Recurrent neural network	4
	Artificial neural network	6
	Hidden Markov model	2
Framework		
	TensorFlow	7
	PyTorch	1
	Scikit learn	1
Programming language		
	Python	8
	MATLAB	5
Performance metrics		
	Accuracy	29
	Kappa Score	5
	Word/character error rates	4
	F1-score	3
	Precision and recall	3
	Sensitivity and specificity	1
Model validation		
	K-fold cross-validation	20
	Train test split	10
	Leave-one-out cross-validation	1
Loss function		
	Cross-entropy	9
	Mean square error	4
	Contrastive loss	1
	Hinge loss	1
	Euclidian, cosine, and correlation distance	1

**Table 3 sensors-22-06975-t003:** Characteristics of Dataset Pre-processing and Feature Extraction.

Characteristics		Study (n)
Dataset Availability		
	Public	13
	Private	21
Dataset Size		
	<500 samples	8
	≥500 & <1000	11
	≥1000 & <1500	4
	≥1500 & <2500	3
	≥2500 & >5000	1
	≥5000	3
Signal Filtering and Normalization		
	Bandpass filter	13
	Bandpass and notch filter	9
	Bandpass and ICA	8
	Kernel principal component analysis (KPCA)	1
	Min–max scaling	1
	Common average reference	1
	Cropped decoding technique, frequency-specific spatial filters	1
Feature extraction		
	Common spatial patterns	5
	Simple features (i.e., min, max, average, std, var, etc.)	5
	Wavelet-vased features	7
	Convolutional neural network	3
	Mel-Cepstral coefficient	3
	Spectral entropy	3
	Covariance-based features	3
	Siamese neural network	1
	Spectrogram	1
	Daubechies	1
	Average power on moving window	1
	Principal representative feature	1
Training set		
	≥90%	11
	≥80%	7
	≥70%	2
	<70%	3
Testing set		
	≥30%	2
	≥20%	8
	<20%	7
Validation set		
	≥20%	2
	<20%	5

**Table 4 sensors-22-06975-t004:** Characteristics of Data Acquisition from EEG Signal.

Characteristics		Study (n)
Subject’s condition		
	Healthy	27
	Not mentioned in the paper	7
Language of the Dataset		
	English	26
	Spanish	4
	Chinese	1
	Korean	1
	Slovak	1
	Persian	1
Prediction labels		
	Words	30
	Sentences	4
Types of speech		
	Covert speech	21
	Overt speech	2
	Both	11
Visual/audio stimuli		
	Yes	33
	No	1
Background noise		
	Yes	7
	No	4
Initial rest		
	≥5 s	8
	3–4 s	4
	1–2 s	9
Rest between trails		
	≥5 s	7
	3–4 s	7
	1–2 s	9
Toolbox for recording		
	EEGLAB	14
	PyEEG	1
	E-Prime	2
	OpenBMI & BBCI	2
	NeuroScan	1
Setting		
	Office	5
	Lab	3
	Isolated room	2
Devices		
	NeuroScan	8
	Brain Products	9
	Emotive	3
	OpenBCI	1
	Biosemi Active Two	1
Number of electrodes		
	128 electrodes	1
	64 electrodes	19
	32 electrodes	4
	Less than 32 electrodes	6

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
