# Peer review of "The Role of Artificial Intelligence in Decoding Speech from EEG Signals: A Scoping Review"

_sensors, 2022, doi:10.3390/s22186975_

Round 1

Reviewer 1 Report

Authors provide the review of EEG signals using AI for BCI systems. In the review, authors showed EEG and AI for decoding speech from EEG signals. The analysis for EEG signal feature extraction and classification based on google scalar, IEEE, and other search engine looks effective and valuable.Discussion section for summary is also useful. Thus, the manuscript could be minor revision after addressing following comments.

1. Figure 1 fonts are not clear to be seen.

2. Please provide ref. with the sentence (significantly impairing one's quality of life and occasionally ~) with ref.(https://www.sciencedirect.com/science/article/abs/pii/S0003682X20306538)

3. In Line 255, there is unnecessary empty space. please remove that.

4. Authors mentioned that BCI researcher used other metrics in Line 281. Is there any reason ?

5. Funding, Conflicts of Interest, and Author Contributions sections are missing.

Author Response

Response to the Reviewers’ Comments: “The Role of Artificial Intelligence in Decoding Speech from EEG Signals: A Scoping Review”

We would like to thank both reviewers for their valuable time and excellent suggestions. In what follows, we provide an item-by-item discussion of how we addressed the reviewers’ suggestions and the requested changes to the manuscript. We believe the manuscript has significantly matured in the process and hope it is ready for publication.

Comments from Reviewer 1:

Authors provide the review of EEG signals using AI for BCI systems. In the review, authors showed EEG and AI for decoding speech from EEG signals. The analysis for EEG signal feature extraction and classification based on google scalar, IEEE, and another search engine looks effective and valuable. Discussion section for summary is also useful. Thus, the manuscript could be minor revision after addressing following comments.

Once again, we thank the reviewer for their constructive comments.

  1. Figure 1 fonts are not clear to be seen.

Response: The figure has been amended.

  1. Please provide ref. with the sentence (significantly impairing one's quality of life and occasionally ~) with ref.(https://www.sciencedirect.com/science/article/abs/pii/S0003682X20306538)

Response: The citation has been added. (Line 38)

  1. In Line 255, there is unnecessary empty space. please remove that.

Response: The unnecessary space has been removed.

  1. Authors mentioned that BCI researcher used other metrics in Line 281. Is there any reason?

Response: We have addressed the reviewer comments by adding the following lines.

We recommend the BCI developer use other performance metrics such as AUC, sensitivity, specificity, and mean square to provide more holistic view of the model performance. It will allow us to understand how the model is sensitive to different classes

  1. Funding, Conflicts of Interest, and Author Contributions sections are missing.

Response: We have addressed the reviewer comments by adding the following lines

Author Contributions: Conceptualization, U.S., A.A.A, and M.A.; data curation, U.S., F.A.M, and M.A.; methodology, U.S. and A.A.A; supervision, M.H., A.A.A and T.A.; validation, A.A.A. and T.A.; writing—original draft, U.S., M.A and F.A.M.; writing—review and editing, U.S., A.A.A, T.A and M.H. All authors have read and agreed to the published version of the manuscript.

Conflict of Interest: The authors declare no conflict of interest.

Funding: This research received no specific grant from any funding agency in the public, commercial, or not-for-profit sectors.”

Reviewer 2 Report

Good work. 

Can you please explain which 34 studies were included in this review from the reference list? 

Thank you.

Author Response

Response to the Reviewers’ Comments: “The Role of Artificial Intelligence in Decoding Speech from EEG Signals: A Scoping Review”

We would like to thank both reviewers for their valuable time and excellent suggestions. In what follows, we provide an item-by-item discussion of how we addressed the reviewers’ suggestions and the requested changes to the manuscript. We believe the manuscript has significantly matured in the process and hope it is ready for publication.

Comments from Reviewer 2:

Good work. 

Once again, we thank the reviewer for their constructive comments

  1. Can you please explain which 34 studies were included in this review from the reference list? 

Response: Please find the list of included studies below.

[2]      K. Mohanchandra and S. Saha, “A communication paradigm using subvocalized speech: translating brain signals into speech,” Augmented Human Research, vol. 1, no. 1, pp. 1–14, 2016.

[3]      M. Koctúrová and J. Juhár, “A Novel Approach to EEG Speech Activity Detection with Visual Stimuli and Mobile BCI,” Applied Sciences, vol. 11, no. 2, p. 674, 2021.

[4]      D.-Y. Lee, M. Lee, and S.-W. Lee, “Decoding Imagined Speech Based on Deep Metric Learning for Intuitive BCI Communication,” IEEE Transactions on Neural Systems and Rehabilitation Engineering, vol. 29, pp. 1363–1374, 2021.

[5]      A. Rezazadeh Sereshkeh, R. Yousefi, A. T. Wong, F. Rudzicz, and T. Chau, “Development of a ternary hybrid fNIRS-EEG brain–computer interface based on imagined speech,” Brain-Computer Interfaces, vol. 6, no. 4, pp. 128–140, 2019.

[8]      M. A. Bakhshali, M. Khademi, A. Ebrahimi-Moghadam, and S. Moghimi, “EEG signal classification of imagined speech based on Riemannian distance of correntropy spectral density,” Biomedical Signal Processing and Control, vol. 59, 2020, [Online]. Available: https://www.scopus.com/inward/record.uri?eid=2-s2.0-85079830821&doi=10.1016%2fj.bspc.2020.101899&partnerID=40&md5=1db1813bbfa9f186b7d6f3a3d0492daf

[9]      A. Balaji et al., “EEG-based classification of bilingual unspoken speech using ANN,” in 2017 39th Annual International Conference of the IEEE Engineering in Medicine and Biology Society (EMBC), 2017, pp. 1022–1025.

[10]     C. Cooney, A. Korik, R. Folli, and D. Coyle, “Evaluation of Hyperparameter Optimization in Machine and Deep Learning Methods for Decoding Imagined Speech EEG.,” Sensors (Basel), vol. 20, no. 16, 2020, [Online]. Available: https://pubmed.ncbi.nlm.nih.gov/32824559/

[11]     A. A. Torres-García, C. A. Reyes-García, L. Villaseñor-Pineda, and G. García-Aguilar, “Implementing a fuzzy inference system in a multi-objective EEG channel selection model for imagined speech classification,” Expert Systems with Applications, vol. 59, pp. 1–12, 2016.

[13]     C. Cooney, R. Folli, and D. Coyle, “Mel Frequency Cepstral Coefficients Enhance Imagined Speech Decoding Accuracy from EEG,” in 2018 29th Irish Signals and Systems Conference (ISSC), 2018, pp. 1–7.

[14]     M. N. I. Qureshi, B. Min, H. Park, D. Cho, W. Choi, and B. Lee, “Multiclass Classification of Word Imagination Speech With Hybrid Connectivity Features,” IEEE Transactions on Biomedical Engineering, vol. 65, no. 10, pp. 2168–2177, 2018.

[15]     D. Pawar and S. Dhage, “Multiclass covert speech classification using extreme learning machine,” Biomedical Engineering Letters, vol. 10, no. 2, pp. 217–226, 2020, [Online]. Available: https://www.scopus.com/inward/record.uri?eid=2-s2.0-85081557774&doi=10.1007%2fs13534-020-00152-x&partnerID=40&md5=ce6dbf7911aee9a8fabad5cf11fbbcfd

[16]     R. A. Sharon, S. Narayanan, M. Sur, and A. Hema Murthy, “Neural Speech Decoding During Audition, Imagination and Production”, doi: 10.1109/ACCESS.2020.3016756.

[17]     N. Hashim, A. Ali, and W.-N. Mohd-Isa, “Word-based classification of imagined speech using EEG,” in International Conference on Computational Science and Technology, 2017, pp. 195–204.

[18]     F. Li et al., “Decoding imagined speech from EEG signals using hybrid-scale spatial-temporal dilated convolution network.,” J Neural Eng, vol. 18, no. 4, 2021, [Online]. Available: https://pubmed.ncbi.nlm.nih.gov/34256357/

[19]     D. Y. Lee, M. Lee, and S. W. Lee, “{Classification of Imagined Speech Using Siamese Neural Network},” IEEE Transactions on Systems, Man, and Cybernetics: Systems, vol. 2020, pp. 2979--2984, 2020.

[20]     C. Cooney, A. Korik, F. Raffaella, and D. Coyle, “Classification of imagined spoken word-pairs using convolutional neural networks,” in The 8th Graz BCI Conference, 2019, 2019, pp. 338–343.

[26]     A. Rezazadeh Sereshkeh, R. Trott, A. Bricout, and T. Chau, “EEG Classification of Covert Speech Using Regularized Neural Networks,” IEEE/ACM Transactions on Audio Speech and Language Processing, vol. 25, no. 12, pp. 2292–2300, Dec. 2017, doi: 10.1109/TASLP.2017.2758164.

[27]     S. H. Lee, M. Lee, and S. W. Lee, “EEG Representations of Spatial and Temporal Features in Imagined Speech and Overt Speech,” Lecture Notes in Computer Science (including subseries Lecture Notes in Artificial Intelligence and Lecture Notes in Bioinformatics), vol. 12047 LNCS, pp. 387–400, 2020, doi: 10.1007/978-3-030-41299-9_30.

[28]     C. H. Nguyen, G. K. Karavas, and P. Artemiadis, “Inferring imagined speech using EEG signals: a new approach using Riemannian manifold features,” Journal of Neural Engineering, vol. 15, no. 1, p. 016002, Dec. 2017, doi: 10.1088/1741-2552/AA8235.

[29]     S.-H. Lee, M. Lee, and S.-W. Lee, “Neural Decoding of Imagined Speech and Visual Imagery as Intuitive Paradigms for BCI Communication,” IEEE Transactions on Neural Systems and Rehabilitation Engineering, vol. 28, no. 12, pp. 2647–2659, 2020.

[30]     G. Krishna, C. Tran, M. Carnahan, and A. Tewfik, “Advancing speech recognition with no speech or with noisy speech,” in 2019 27th European Signal Processing Conference (EUSIPCO), 2019, pp. 1–5.

[31]     S. Zhao and F. Rudzicz, “Classifying phonological categories in imagined and articulated speech,” in 2015 IEEE International Conference on Acoustics, Speech and Signal Processing (ICASSP), 2015, pp. 992–996.

[32]     A.-L. Rusnac and O. Grigore, “Convolutional Neural Network applied in EEG imagined phoneme recognition system,” in 2021 12th International Symposium on Advanced Topics in Electrical Engineering (ATEE), 2021, pp. 1–4.

[33]     R. A. Sharon and H. A. Murthy, “Correlation based Multi-phasal models for improved imagined speech EEG recognition,” arXiv preprint arXiv:2011.02195, 2020.

[34]     J. T. Panachakel, A. G. Ramakrishnan, and T. v Ananthapadmanabha, “Decoding Imagined Speech using Wavelet Features and Deep Neural Networks,” in 2019 IEEE 16th India Council International Conference (INDICON), 2019, pp. 1–4.

[35]     P. Saha, S. Fels, and M. Abdul-Mageed, “Deep Learning the EEG Manifold for Phonological Categorization from Active Thoughts,” in ICASSP 2019 - 2019 IEEE International Conference on Acoustics, Speech and Signal Processing (ICASSP), 2019, pp. 2762–2766.

[36]     M. M. Islam and M. M. H. Shuvo, “DenseNet Based Speech Imagery EEG Signal Classification using Gramian Angular Field,” in 2019 5th International Conference on Advances in Electrical Engineering (ICAEE), 2019, pp. 149–154.

[37]     M. Alsaleh, R. Moore, H. Christensen, and M. Arvaneh, “Examining Temporal Variations in Recognizing Unspoken Words Using EEG Signals,” in 2018 IEEE International Conference on Systems, Man, and Cybernetics (SMC), 2018, pp. 976–981.

[38]     N. Hamedi, S. Samiei, M. Delrobaei, and A. Khadem, “Imagined Speech Decoding From EEG: The Winner of 3rd Iranian BCI Competition (iBCIC2020),” in 2020 27th National and 5th International Iranian Conference on Biomedical Engineering (ICBME), 2020, pp. 101–105.

[39]     T. Kim, J. Lee, H. Choi, H. Lee, I. Y. Kim, and D. P. Jang, “Meaning based covert speech classification for brain-computer interface based on electroencephalography,” International IEEE/EMBS Conference on Neural Engineering, NER, pp. 53–56, 2013, doi: 10.1109/NER.2013.6695869.

[40]     B. H. Lee, B. H. Kwon, D. Y. Lee, and J. H. Jeong, “{Speech Imagery Classification using Length-Wise Training based on Deep Learning},” 9th IEEE International Winter Conference on Brain-Computer Interface, BCI 2021, 2021.

[41]     G. Krishna, C. Tran, M. Carnahan, and A. Tewfik, “Continuous Silent Speech Recognition using EEG,” arXiv preprint arXiv:2002.03851, 2020.

[42]     G. Krishna, Y. Han, C. Tran, M. Carnahan, and A. H. Tewfik, “State-of-the-art speech recognition using eeg and towards decoding of speech spectrum from eeg,” arXiv preprint arXiv:1908.05743, 2019.

[43]     S.-H. Lee, Y.-E. Lee, and S.-W. Lee, “{Voice of Your Brain: Cognitive Representations of Imagined Speech,Overt Speech, and Speech Perception Based on EEG},” pp. 2--6, 2021, [Online]. Available: http://arxiv.org/abs/2105.14787

Round 2

Reviewer 1 Report

Authors revised the manuscript very well so I recommend this article without further revision anymore to be accepted as it is.

Reviewer 2 Report

Thank you for submitting the revised work.